# Human Macrophages Polarized by Interaction with Apoptotic Cells Produce Fibrosis-Associated Mediators and Enhance Pro-Fibrotic Activity of Dermal Fibroblasts In Vitro

**DOI:** 10.3390/cells12151928

**Published:** 2023-07-25

**Authors:** Aleksandra Maksimova, Ekaterina Shevela, Lyudmila Sakhno, Marina Tikhonova, Aleksandr Ostanin, Elena Chernykh

**Affiliations:** Research Institute of Fundamental and Clinical Immunology, Novosibirsk 630099, Russia; shevelak@mail.ru (E.S.); lsahno53@mail.ru (L.S.); martix-59@mail.ru (M.T.); ostanin62@mail.ru (A.O.); ct_lab@mail.ru (E.C.)

**Keywords:** macrophage, fibroblast, efferocytosis, repair, wound healing, collagen, matrix metalloproteinase

## Abstract

Apoptosis and subsequent removal of dead cells are an essential part of wound healing. Macrophages phagocytize apoptotic cells (efferocytosis) and contribute to the resolution of inflammation. However, their participation in fibrogenesis and the mechanisms of influence on this process remain unclear. In the present study, we focused on the fibrogenic properties of human monocyte-derived macrophages polarized in the M2 direction by interaction with apoptotic cells. We studied their influence on the proliferation ([3H]-thymidine incorporation), differentiation (by the expression of α-SMA, a myofibroblast marker) and collagen-producing activity (ELISA) of dermal fibroblasts compared to classically (LPS) and alternatively (IL-4) activated macrophages. Macrophages polarized by the interaction with apoptotic cells had a unique phenotype and profile of produced factors and differed from the compared macrophage subtypes. Their conditioned media promoted the proliferation of dermal fibroblasts and the expression of α-SMA in them at the level of macrophages stimulated by IL-4, while the stimulating effect on the collagen-producing activity was more pronounced compared to that of the other macrophage subtypes. Moreover, they are characterized by the high level of production of pro-fibrotic factors such as TIMP-1, TGF-β1 and angiogenin. Taken together, M2-like macrophages polarized by efferocytosis demonstrate in vitro pro-fibrotic activity by promoting the functional activity of dermal fibroblasts and producing pro-fibrotic and pro-angiogenic factors.

## 1. Introduction

The macrophage ability to regulate fibroblast functions and extracellular matrix (ECM) turnover through direct and/or indirect pathways underlies their essential role in tissue repair and fibrosis [1,2]. Reparative response to tissue damage triggers a series of dynamic processes including inflammation, proliferation and tissue remodeling [3]. The self-limiting inflammatory response is necessary to protect damaged tissue from pathogens and initiate the proliferation phase that in most tissues is implemented by the formation of new granulation tissue. In the proliferative phase, fibroblasts proliferate and migrate to the damage site, whereupon some of them differentiate into myofibroblasts and produce new ECM proteins, mainly collagen. Alpha-smooth muscle actin (α-SMA) being the specific marker of myofibroblasts [4] provides these cells with the ability to contractile activity, increased mobility and synthesis/production of ECM [5]. In the remodeling phase, the apoptosis of the myofibroblasts and reorganization of ECM result in the transformation of granulation tissue into scar tissue.

Most authors now agree that different macrophage subtypes influence differently at each stage of wound healing using many mechanisms [6,7]. The direct effect on ECM turnover is primarily associated with the production of matrix metalloproteinases (MMPs) and their tissue inhibitors (TIMPs) by macrophages [8,9], as well as the expression of surface receptors responsible for the phagocytosis of degraded or intact collagen [10,11]. In turn, macrophages can affect fibrogenesis indirectly through the stimulation and activation of fibroblasts via the contact- and cytokine-dependent pathway [6].

The multiple biological activities macrophages perform are currently explained by cell plasticity. During the reparative response macrophages can adopt various functional phenotypes depending on microenvironment stimuli [12]. In vitro studies have demonstrated that Th1 and Th2 cytokines can differentially polarize macrophages into classically activated (M1) and alternatively activated (M2) ones. The M1 macrophages exert pro-inflammatory and antimicrobial activities, while the M2 macrophages possess anti-inflammatory and reparative activities [13]. Further, according to their response to different modulators, the M2 macrophages are divided into M2a, M2b, M2c and M2d phenotypes, which differ in their cell surface markers, secreted cytokines and biological functions [14,15,16].

Apoptosis and subsequent removal of dead cells is an essential part of fibrogenesis and a key regulatory mechanism for limiting inflammation and initiating healing [17,18]. A high level of apoptosis has been noted in almost all types of fibrosis, and many mechanisms have been proposed by which apoptotic cells can determine the outcomes of fibrosis [19]. Macrophages phagocytizing apoptotic cells (efferocytosis) are considered to be key cells contributing to the resolution of inflammation [18]. Such macrophages are characterized by lower levels of TNFα and IL-6 secretion and high levels of IL-10, TGF-β1 and IGF-1, and it is assumed that these macrophages are able to initiate a pro-fibrotic response promoting fibroblasts proliferation and differentiation into myofibroblasts [18]. For example, it has been shown that in mice, the impairment of phagocytosis of apoptotic neutrophils by macrophages is associated with a decrease in TGF-β1 production, a decrease in the number of myofibroblasts and is accompanied by a slowdown in wound healing [20]. Nevertheless, the question remains whether and how efferocytosis affects the fibroblast-modulating activity of human macrophages. Previously, we developed the original protocol for obtaining M2-like macrophages generated from human blood monocytes under low-serum conditions designated as the M2(LS, low-serum) macrophages [21]. In this protocol, macrophage polarization toward the M2 phenotype is the result of interaction with apoptotic cells induced by serum deprivation. However, the fibroblast-modulating properties of the M2(LS) macrophages have not been studied before. Given the critical importance of apoptosis and further phagocytic clearance of apoptotic cells during wound healing, we studied the M2(LS) macrophages’ ability to produce fibrosis-associated factors and their influence on the proliferation, differentiation and collagen-producing activity of primary dermal fibroblasts compared to classically and alternatively activated human macrophages.

## 2. Materials and Methods

This study included 72 healthy donors aged 22–60. Informed consent was obtained following the Declaration of Helsinki. All the experiments using human samples were performed according to a protocol approved by the Institutional Review Board of the Research Institute of Fundamental and Clinical Immunology (protocol №123, 4 June 2021). M1 and M2 macrophages were generated as described in [22]. Briefly, monocytes were isolated from heparinized venous blood by the adhesion method and then cultured on 12-well plates (TPP, Trasadingen, Switzerland) in the presence of 50 ng/mL recombinant GM-CSF (Sigma-Aldrich, Burlington, NJ, USA) for 7 days. On the 5th day, appropriate polarizing stimuli were added to the cultures: 10 μg/mL lipopolysaccharide, LPS (*E. coli* 0114: B4, Sigma-Aldrich, Burlington, NJ, USA), to obtain M1; 20 ng/mL IL-4 (Sigma-Aldrich, Burlington, NJ, USA) to obtain M2. To obtain M2(LS), the protocol of [23] was used. Briefly, a medium supplemented with GM-CSF (50 ng/mL) and 2% autoplasma (conditions of deficiency of growth/serum factors) was used. The adhesion time was increased to 18 h, which is necessary and sufficient for the induction of apoptosis in the fraction of nonadhesive mononuclear cells and the engulfment of apoptotic cells by monocytes. After 7 days, macrophages were gently scraped off, counted, and their viability was determined (excluding trypan blue). Supernatants were collected, centrifuged, cryopreserved and stored at −80 °C.

The source of fibroblasts was the dermal fibroblast cell line NAF1 obtained from the skin of a burn patient. The cell line was kindly provided by the Center for Collective Use “Collection of Pluripotent Human and Mammalian Cell Cultures of General Biological and Biomedical Direction” of the Federal Research Center Institute of Cytology and Genetics. Cells were passaged at approximately 70% confluence.

To study the functional activity of fibroblasts (differentiation into myofibroblasts and collagen production), NAF1 cells were cultured in a conditioned medium of various macrophages for 24 h, which was then replaced with a serum-free DMEM/F12 medium (Biolot, Saint Petersburg, Russia) for 4 days.

The concentration of cytokines and chemokines was assessed in 7-day macrophage supernatants using the Bio-Plex Pro Human Cytokine Grp I Panel 8- and 17-Plex test systems (Bio-Rad, Hercules, CA, USA), following the manufacturer’s instructions.

To determine the concentration of VEGF, TGF-β1, angiogenin (ribonuclease 5), MMP-9, MMP-2 and TIMP-1 in the supernatants of 7-day macrophage cultures, we used the appropriate ELISA kit (all kits of the R&D System, Minneapolis, USA) according to the manufacturer’s instructions. Before assaying the concentration of TGF-β1, the latent form of the factor in the test samples was converted to the active form following the guidelines.

The level of collagen I production was evaluated as the concentration of the α1-chain of collagen I in supernatants of fibroblasts treated with macrophage-conditioned media using Human COL1A1 (Collagen Type I Alpha 1) ELISA Kit (FineTest, Wuhan, China) following the manufacturer’s instruction.

The allostimulatory activity of macrophages was determined by measuring allogeneic T cell proliferation in the mixed leukocyte culture as described in [22]. Briefly, peripheral blood mononuclear cells (PBMC) in an amount of 1 × 10^5^/well were plated on 96-well tissue culture plates in the presence of different macrophage subtypes (1 × 10^4^/well). Proliferation of allogeneic T cells was determined radiometrically by [3H]-thymidine incorporation. The allostimulatory activity of macrophages was expressed by a stimulation index calculated as a ratio of PBMC proliferation in the presence of macrophages to spontaneous PBMC proliferative responses.

The proliferative response of dermal fibroblasts of the NAF1 line was determined by the incorporation of [3H]-thymidine. For this, NAF1 cells were cultured on a 96-well plate (TPP, Trasadingen, Switzerland) at a concentration of 5 × 10^3^ cells/well in the conditioned media of various macrophage subtypes or DMEM/F12 medium (Biolot, Saint Petersburg, Russia) (spontaneous proliferation) for 24 h; then, the conditioned media were removed, the cells were washed once and further cultivated in DMEM/F12 medium for up to 5 days. The [3H]-thymidine was added 18 h before the end of cultivation (1 μCu/well). The index of stimulation of proliferation of fibroblasts was calculated as the ratio of the proliferative response of fibroblasts treated with the macrophage-conditioned media to the level of spontaneous proliferation (negative control).

The macrophage phenotype was evaluated by flow cytometry [21]. Polarized macrophages were stained using fluorochrome-tagged monoclonal antibodies (CD14-FITC, CD86-FITC, -HLA-DR-PE, CD206-PE, CD163-PerCP, MerTK-AlexaFluor647) (all from BD PharMingen, San Diego, CA, USA). Samples were analyzed using FACSCalibur (Becton Dickinson, San Diego, CA, USA) and the Cell Quest program (Becton Dickinson, San Diego, CA, USA), and the percentage of positive cells expressing the corresponding markers was determined.

To assess the intracellular expression of alpha-smooth muscle actin (α-SMA), fibroblasts were trypsinized, collected, washed with PBS, treated with permeabilizing solutions (Transcription Factor Buffer Set, BD PharMingen, San Diego, CA, USA) followed by treatment with APC-conjugated α-SMA antibodies (R&D Systems, Minneapolis, MI, USA). The TGF-β1-induced α-SMA expression (PeproTECH, Cranbury, NJ, USA) was a positive control. Spontaneous differentiation of fibroblasts into myofibroblasts in a serum-free medium was a negative control. Samples were analyzed using a FACSCalibur (Becton Dickinson, Franklin Lakes, NJ, USA) and the Cell Quest program (Becton Dickinson, Franklin Lakes, NJ, USA), and the percentage of positive cells expressing the corresponding markers was determined. Visualization was performed using FCS Express Version 3.

The results were statistically processed using the STATISTICA 8.0 software (StatSoft. Inc., Oklahoma, OK, USA). Data are presented as median with the indication of interquartile ranges (Me, IQR) and minimum and maximum. The significance of the differences between the compared groups was assessed using the Wilcoxon matched-pair test; differences were considered significant at *p* < 0.05.

## 3. Results

### 3.1. Characteristic of Polarized Macrophages

To characterize the M2(LS) macrophages, we first assessed the phenotype of these cells and their production of pro- and anti-inflammatory cytokines. As comparison macrophage subsets, we used classically activated macrophages induced by LPS, M1(LPS) and alternatively activated macrophages induced by IL-4, M2(IL-4). First, the studied macrophages were characterized by the expression of the cell surface molecules, including HLA-DR, CD14, CD86, CD163, CD206 and Mer tyrosine kinase (MerTK) (Figure 1 and Appendix B). Despite significant inter-donor variability in marker expression (Figure 1), the M1(LPS) cultures differed by a higher number of CD86-expressing cells, while the M2(IL-4) cultures were characterized by high levels of the CD163-, CD206- and MerTK-expressing cells. The phenotype of the M2(LS) macrophages was closer to that of the M2(IL-4) cells than to the phenotype of the M1(LPS) cells, since they differed in a higher content of CD206+ and CD163+ cells (*p* = 0.06 and 0.04, respectively) and fewer CD86+ cells (*p* = 0.04) compared to the M1(LPS) macrophages. Of note, the content of the CD206+ cells in the M2(LS) cultures was higher than that in the M2(IL-4) cultures (*p* = 0.04), while differences in the CD163- and MerTK-expressing cells were not significant.

Secretome analysis showed that similar to the M2(IL-4) cells, the M2(LS) macrophages were characterized by significantly lower pro-inflammatory (IL-1, TNFα, CCL3 or MIP-1β) and immunoregulatory (IL-2, -5, -6, -12 and -17) cytokine levels compared to those of the M1(LPS) macrophages (Figure 2 and Appendix A). In addition, we also found lower levels of IL-4 and IL-10 in the M2(IL-4) and M2(LS) supernatants compared to those of the M1(LPS) macrophages. However, the concentration of IL-1, -5, -6 and -17 in the M2(LS) supernatants exceeded that in the M2(IL-4) supernatants (*p* < 0.05), although it did not reach the level of the M1 macrophages.

One of the distinguishing features of the M2 macrophages is their low allostimulatory activity, i.e., the ability to enhance the proliferation of allogeneic T cells in a mixed leukocyte culture [22]. Figure 3 shows that the M2(LS), as well as the M2(IL-4) macrophages, were characterized by significantly lower stimulatory indices compared to that of the M1(LPS) macrophages—2.3 (both) vs. 9.3, respectively (*p* = 0.00065 and *p* = 0.000006).

Taken together, the M2(LS) macrophages being similar to the M2(IL-4) cells by low allostimulatory activity are different in terms of a unique phenotype and produced cytokines.

### 3.2. Characteristic of Polarized Macrophages

In the next stage, we evaluated the effect of the macrophage-conditioned media on the functional activity of dermal fibroblasts. The study of fibrogenesis in in vivo mouse models failed to clarify the precise mechanisms of fibroblast regulation by macrophages, since many pro- and antifibrotic effects observed could be mediated through indirect paracrine mechanisms [7]. In this case, in vitro approaches help to define better whether and how differently activated macrophages regulate fibroblast functions. Given that the formation and remodeling of granulation tissue implement fibroblast proliferation and differentiation and the synthesis of the collagen matrix, the evaluation of fibroblast functions included the detection of all these parameters.

First of all, we studied the influence of the macrophages on the fibroblast proliferative activity. As shown in Figure 4a, the M2(LS)-conditioned medium increased the fibroblast proliferation by more than three times (*p* = 0.043, compared to the control). The M2(LS) effect did not differ from that of the M2(IL-4) (median 3.58) and slightly exceeded the stimulating activity of the M1(LPS) macrophages (2.3, *p* = 0.07).

To study fibroblast differentiation, we determined the expression of α-SMA as the main marker of myofibroblasts (Figure 4b). The M2(LS)-conditioned medium increased the number of α-SMA+ cells in the fibroblast cultures from 33% to 58% (*p* = 0.028, compared to the control). The M2(IL-4) supernatants demonstrated a similar stimulating effect (an increase in α-SMA+ cells from 33% to 62%; *p* = 0.046). In contrast, the M1(LPS) supernatants did not have such an effect (48% of α-SMA+ cells; *p* = 0.28, compared to the control). The pro-differentiated effect of the M2(LS)-conditioned medium was slightly lower than that of the TGF-β, which was used as a positive control (*p* = 0.04).

We also assessed the capacity of fibroblasts to produce collagen I, which is the predominant component of ECM in the skin. Figure 4c shows that the conditioned media of all macrophage subtypes increased collagen production by dermal fibroblasts. The highest concentration of collagen (107.3 ng/mL) was determined in the cultures of fibroblasts pretreated with the M2(LS) supernatants. Thus, the collagen-stimulating activity of the M2(LS) macrophages exceeded that of the M1(LPS) (77.3 ng/mL, *p* = 0.06) and M2(IL-4) (72.7 ng/mL, *p* = 0.04) macrophages. The concentration of collagen I in the cultures pretreated with the conditioned medium of the M2(LS) macrophages was similar to the TGF-β-induced collagen production (101.3 ng/mL).

### 3.3. MMP and TIMP Production

MMPs and TIMPs are the most important mediators of ECM turnover. Therefore, we assessed the production of MMP-2, MMP-9 and TIMP-1, which are known to be produced by human macrophages. There were no significant differences between the different cultures of macrophages in both MMP-2 and MMP-9 production. Noteworthy, the levels of MMP-2 production by all macrophage subsets were lower compared to MMP-9 (on average 33.5–48.0 and 6600–6800 pg/mL, respectively) (Figure 5a,b).

On the contrary, the M2(LS) macrophages greatly differed from the M1 and M2 macrophages by the TIMP-1 production (Figure 5c). The concentration of TIMP-1 in the M2(LS) supernatants was 4300 (IQR 2600–5800) pg/mL and significantly exceeded that in the M2(IL-4) supernatants (2100, *p* = 0.005) and as a trend in the M1(LPS) cultures (3100, *p* = 0.07).

We also evaluated the MMP-to-TIMP ratio (Figure 5d,e) and found that the MMP-2/TIMP-1 ratio for the M2(LS) macrophages was significantly lower than that for the M2(IL-4) macrophages (0.01 vs. 0.1 *p* = 0.04) due to the higher content of TIMP-1. Similarly, the MMP-9/TIMP-1 ratio for the M2(LS) macrophages was significantly lower compared to that for the M2(IL-4) macrophages (3.28 vs. 9.0, *p* = 0.04).

Thus, the M2(LS) cells can be characterized as more conducive to ECM accumulation due to the higher production of TIMPs.

### 3.4. Production of TGF-β, VEGF and Angiogenin

Finally, we evaluated the production of macrophage-derived factors involved in the regulation of fibrogenesis, in particular, TGF-β, VEGF and angiogenin. Among the macrophage subsets studied, the M2(LS) macrophages demonstrated the highest production of TGF-β, which is considered a key factor of fibroblast differentiation (Figure 6a). The TGF-β concentration in the M2(LS) supernatants reached 9200 pg/mL and was approximately two times higher compared to that in the M1(LPS) and M2(IL-4) supernatants (*p* = 0.06 and 0.04, respectively).

As shown in Figure 6b, the M2(LS) macrophages also actively secreted VEGF, which has pronounced fibroblast-modulating activity along with pro-angiogenic properties. The concentration of VEGF in the M2(LS) cultures did not differ from its content in the M2(IL-4) supernatants (2600 pg/mL and 2200 pg/mL, respectively) and exceeded the level of this factor in the M1(LPS) cultures as a trend (320 pg/mL, *p* =0.07).

In addition, the M2(LS) macrophages demonstrated the highest production of angiogenin. It is another pro-angiogenic factor, which plays an important role in fibrogenesis. In particular, the concentration of angiogenin in the M2(LS) cultures was more than four times higher than that in the M1(LPS) and M2(IL-4) supernatants (4700 pg/mL vs. 1700 pg/mL and 1200 pg/mL, respectively; *p* < 0.05) (Figure 6c).

Thus, the polarization via the interaction with apoptotic cells induces the activation of the macrophages that more actively produce pro-fibrogenic factors compared to the LPS- or IL-4-activated macrophages.

## 4. Discussion

In the current work, we studied human macrophages polarized towards M2 phenotype through the interaction with apoptotic cells (M2(LS)), in particular, the direct capacity to modulate fibroblast functions and the patterns of fibrosis-related factors in comparison with macrophages activated with traditional M1 or M2 stimuli (LPS or IL-4). Given that the reparative response is initiated by inflammation, macrophages were differentiated from blood monocytes in the presence of GM-CSF to modulate the pro-inflammatory microenvironment.

Firstly, the data obtained showed that the M2(LS) macrophages possessed features of an M2-like phenotype, since they expressed prototypic M2 markers (CD163, CD206 and MerTK), had low allostimulatory activity characteristic for the M2 phenotype [22] and produced a high level of M2-associated VEGF and low levels of pro-inflammatory cytokines (IL-1β, TNFα, IL-2, IL-12, IL-6, IL-17 and MIP-1β) as compared with the M1(LPS) macrophages. On the other hand, M2(LS) differed from classical M2(IL-4) by a relatively higher level of IL-1β, IL-6, IL-17 and IL-5, evidencing their unique functional phenotype.

At present, most of the authors are inclined to believe that the interaction of macrophages with apoptotic cells induces an M2-like anti-inflammatory phenotype [24,25,26]. The engulfment of apoptotic cells induces multiple signaling pathways that result in the downregulation of pro-inflammatory cytokines and upregulation of anti-inflammatory and pro-resolving lipid mediators [27,28]. In addition, ATP released from apoptotic cells is converted to adenosine and through the adenosine receptors suppresses the production of pro-inflammatory mediators and chemokines [29]. The uptake of apoptotic cells also activates the autophagy pathway, which is not only involved in the clearance of apoptotic material but also prevents the inflammation-reducing IL-1β and IL-18 production [30]. On the other hand, it has recently been shown that the engulfment of apoptotic endothelial cells in vitro and in vivo leads to the generation of macrophages which have characteristics of both the M1 and M2 macrophage phenotypes [31]. Some earlier studies have also found the M1 features in the macrophages generated as a result of efferocytosis [32,33]. Here, we provide evidence that macrophages polarized by the interaction with apoptotic cells from a nonadherent fraction of mononuclear cells have a unique phenotype and profile of produced factors and differ from the macrophages activated by LPS or IL-4.

Macrophages are critically involved in all phases of tissue repair regulating inflammatory response, clearing cell debris and modulating fibroblast behavior. Immediately after injury, macrophages produce pro-inflammatory cytokines, chemokines, MMPs and other factors that coordinate inflammatory response [6]. Subsequently, macrophage population shifts to reparative and anti-inflammatory/resolving phenotypes that promote the resolution of inflammation and are involved in tissue repair and remodeling. Nevertheless, the capacity of differently activated macrophages to directly modulate fibroblast functions has not been fully characterized.

One of the in vitro approaches to answer the questions whether and how human primary macrophages regulate fibroblast functions, and if the functional phenotype of macrophages determines their fibroblast-modulating activity, is the evaluation of macrophage effects on fibroblast proliferation, differentiation and collagen-producing activity. However, to date only a few studies have been carried out in this field [33,34,35]. These studies provided evidence that spontaneously or M-CSF-differentiated monocyte-derived macrophages can modulate fibroblast functions via the cell-to-cell contact and soluble factors. In addition, they demonstrated that alternatively activated macrophages (polarized with IL-4/IL-13; M2a) generally activated the fibrogenic activity of fibroblasts, whereas classically activated macrophages (stimulated with LPS/IFN-γ) failed to enhance fibroblast proliferation and differentiation [34,35,36].

In our study, GM-CSF-differentiated macrophages with both the M1 and M2 phenotype enhanced fibroblast proliferation and collagen I production, while only the M2 cells could increase fibroblast differentiation. These results coincide with the data of Glim et al. who demonstrated the pro-differentiated effect of the M2(IL4) macrophages and failed to show the same activity of the M1(LPS) macrophages [36]. In contrast, Ploeger et al. reported that both the M1(LPS/IFN-γ)- and M2(IL4/IL-13)-induced fibroblast differentiation with a more pronounced effect of the M2 cells. They also found that only the M2, but not the M1 cells, could stimulate collagen I production [35]. These contradictions may be related to the utilization of M-CSF (instead of GM-CSF in our study) and different polarizing stimuli.

Of note, we firstly demonstrated that M2(LS) polarized via the interaction with apoptotic cells did not differ from M2(IL-4) in their stimulatory effects on fibroblast proliferation and differentiation. However, these cells exceed M2(IL-4) in terms of collagen-producing activity. The results obtained are generally consistent with the study of Nacu et al. who demonstrated that monocyte-derived macrophages following ingestion of apoptotic cells upregulate the collagen production by fibroblasts [37]. In contrast, Kim et al. found the opposite effect: the phagocytosis of apoptotic cells by macrophages led to the formation of cells that decreased the expression of collagen I mRNA and the expression of α-SMA in lung fibroblasts activated by TGF-β1 [38]. These differences may be due to different sources of macrophages (human or mouse) and fibroblasts (skin or lung).

The pro-fibrotic activity of macrophages is known to be mediated by various factors, among which, several cytokines (TGF-β, IL-13, IL-4, IL-6), growth and angiogenic factors (FGF, EGF, PDGF, VEGF, angiogenin) and MMPs and their tissue inhibitors can directly stimulate fibroblast functions, induce or promote fibroblast recruiting and migration or activate other cells to produce pro-fibrotic cytokines [6,39,40]. Therefore, finally we compared the M2(LS) macrophages with traditionally polarized M1 and M2 macrophages in their capacity to produce some key pro-fibrotic factors.

Firstly, we found that all tested macrophages produced MMP-2 and MMP-9, and M2(LS) did not differ from the M1(LPS) and M2(IL-4) macrophages in the levels of these MMPs while being characterized by a higher level of TIMP-1. Human macrophages produce many types of MMPs including MMP-2 and MMP-9 which are two of the most extensively studied members of the MMPs family [41]. These MMPs contribute to the remodeling of the ECM. However, the data available in the scientific literature concerning the association of the MMP production with the stage of maturation/differentiation and the polarization state of macrophages are often inconsistent. Some authors point to the predominance of production of these MMPs in the M1 macrophages compared to M2 [42,43], while others report more prominent production of MMPs by the M2c cells [44]. We did not find significant differences in the MMP levels among differently activated macrophages, which is probably related to the different protocols of macrophage generation. At the same time, we found that the M2(LS) macrophages are characterized by high levels of TIMP-1. The TIMP-1 inhibits the functional activity of MMPs and has pro-proliferative and antiapoptotic effects on fibroblasts [45,46,47]. However, in our study, the high levels of TIMP-1 in the M2(LS) supernatants did not result in a higher capacity to stimulate the fibroblast proliferation.

Secondly, our data show that the level of TGF-β production by the M2(LS) macrophages significantly exceeded that of the LPS- or IL-4-activated macrophages. Previously, it was shown that the phagocytosis of apoptotic cells increases the expression of mRNA of this factor in macrophages [37,48], which is consistent with our results. TGF-β is a powerful pro-fibrotic factor that is thought to be a driver of fibrosis. It promotes the proliferation/differentiation of fibroblasts and stimulates ECM protein production [49,50]. Meanwhile, we observed a link between the TGF-β concentration and the stimulatory activity of the M2(LS) supernatants only with regard to the collagen production but not to the proliferation and α-SMA expression. We suggest that it can be related to the action of other important growth factors capable of regulating fibroblast functions, in particular, PDGF-CC [36].

Finally, M2(LS) were characterized by a higher level of VEGF and especially of angiogenin compared to the other types of macrophages. VEGF is an important pro-angiogenic factor involved in the regeneration. VEGF has an important role in the fibrosis development promoting collagen I and collagen 3 synthesis, proliferation rate and migration capacity, as well as the differentiation of human fibroblasts [51,52]. Currently, M2 macrophages appear to be more effective producers of VEGF than the M1 cells [53] that are generally in agreement with our results. High production of VEGF by M2(LS) macrophages polarized by the interaction with apoptotic cells seems to be expected since efferocytosis triggers the VEGF production [54].

As for the angiogenin production, we were the first to characterize its production by differently activated macrophages and revealed that the M2(LS) macrophages polarized by efferocytosis are the main producers of this multifunctional factor with proangiogenic activity. The main targets of the angiogenin along with the endothelial cells and smooth muscle cells are fibroblasts. The functional activity of the angiogenin substantially determines the course and resolution of the wound healing, directly, by stimulating neovascularization, and indirectly, by activating fibroblasts and the factors they produce. In animals treated with the angiogenin, a significant increase in the number of fibroblasts and the densities of collagen fibers was detected. Therefore, we conclude that the highest capacity of the M2(LS) macrophages to increase collagen production may be partially related to high angiogenin production.

## 5. Conclusions

The data obtained generally suggest that the M2(LS) being M2-like macrophages demonstrate in vitro pro-fibrotic activity by promoting proliferation, differentiation and collagen synthesis by dermal fibroblasts and therefore may be considered as promising candidates for cell therapy aimed at enhancing skin regeneration.

## Figures and Tables

**Figure 1 cells-12-01928-f001:**
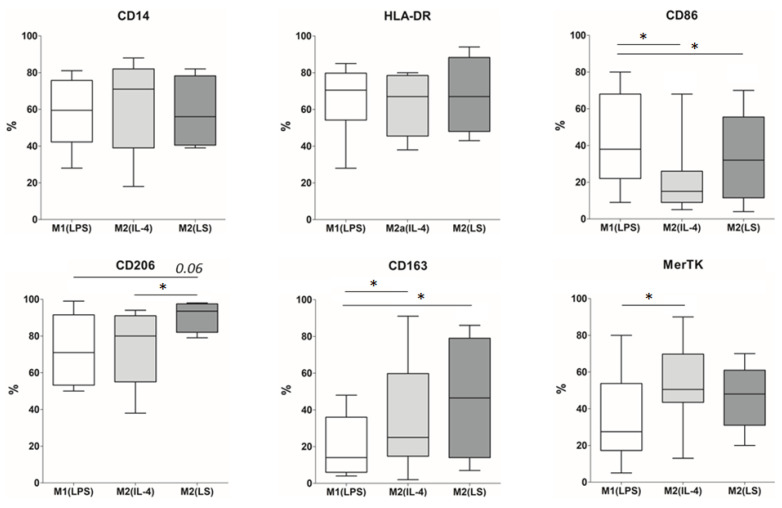
The number of cells (%) expressing macrophage markers in cultures of M1(LPS), M2(IL-4) and M2(LS). Data are presented as median, interquartile range and minimum and maximum; *n* = 4–10; * significant at <0.05.

**Figure 2 cells-12-01928-f002:**
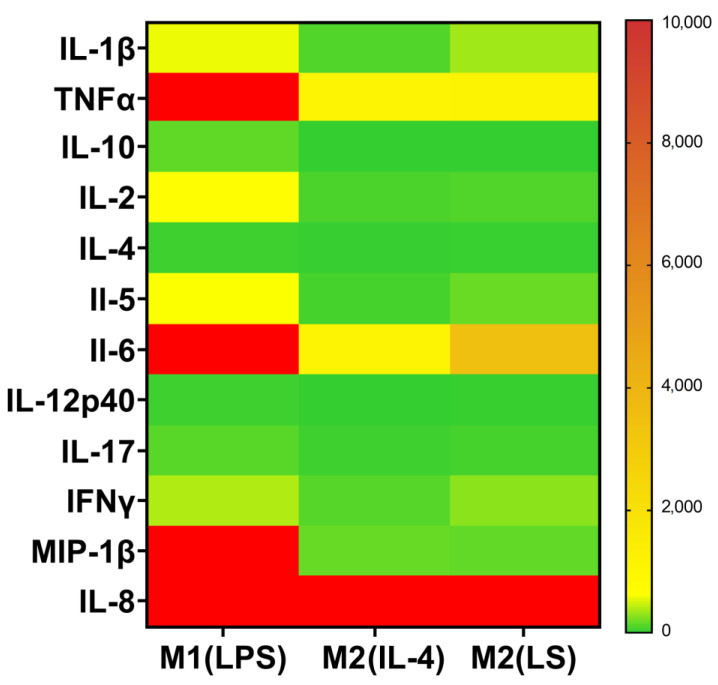
Heat map of cytokine and chemokine concentration (median) in the M1(LPS), M2(IL-4) and M2(LS) supernatants (*n* = 6–16).

**Figure 3 cells-12-01928-f003:**
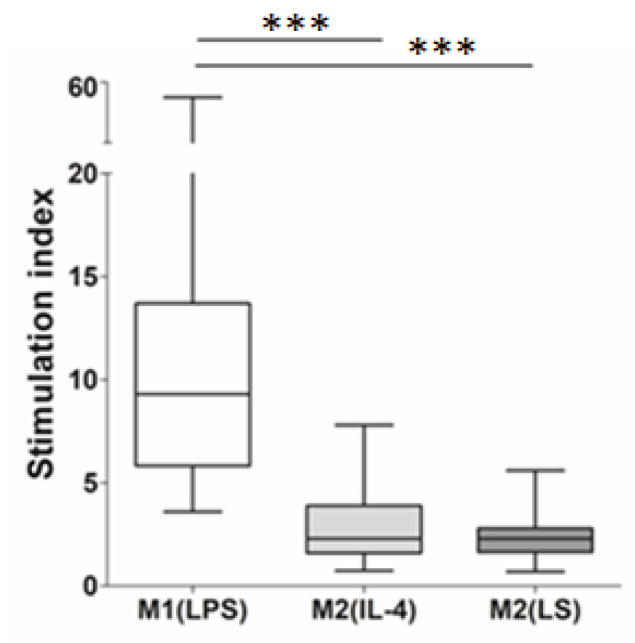
Allostimulatory activity of M1(LPS), M2(IL-4) and M2(LS) macrophages. Data are expressed as stimulation indices and are presented as median, interquartile range and minimum and maximum; *n* = 15–41. *** significant at <0.001.

**Figure 4 cells-12-01928-f004:**
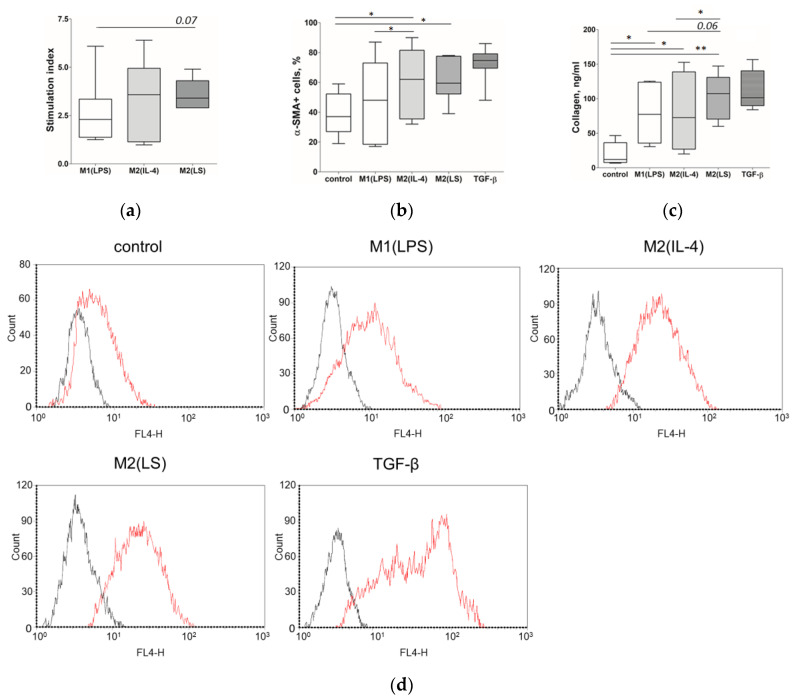
Influence of M1(LPS)-, M2(IL-4)- and M2(LS)-conditioned media on the fibroblast (NAF1) functions. Data are presented as median, interquartile range and minimum and maximum. * significant at <0.05, ** significant at <0.01. (**a**) Proliferation of fibroblasts, expressed as stimulation indices (*n* = 9). (**b**) The content of α-SMA+ fibroblasts in cultures (*n* = 10). (**c**) The concentration of collagen I in the fibroblast supernatants (*n* = 8). (**d**) The expression of α-SMA in fibroblast cultures, flow cytometry data of a representative experiment.

**Figure 5 cells-12-01928-f005:**
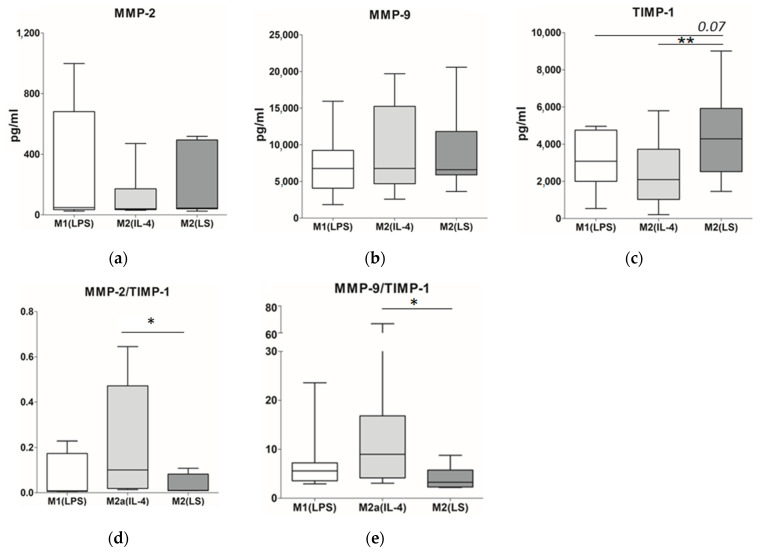
MMP and TIMP production by M1(LPS), M2(IL-4) and M2(LS) macrophages. Data are presented as median, interquartile range and minimum and maximum. * significant at <0.05, ** significant at <0.01. (**a**) MMP-2 production (*n* = 8). (**b**) MMP-9 production (*n* = 11–14). (**c**) TIMP-1 production (*n* = 10–12). (**d**) MMP-2/TIMP-1 ratio (*n* = 5). (**e**) MMP-9/TIMP-1 ratio (*n* = 7–9).

**Figure 6 cells-12-01928-f006:**
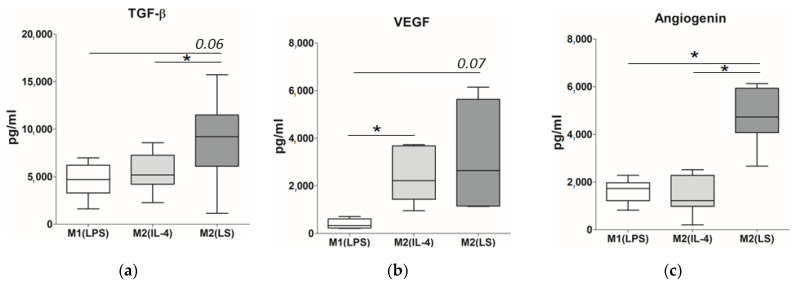
Production of factors involved in the fibrogenesis regulation by M1(LPS), M2(IL-4) and M2(LS) macrophages. Data are presented as median, interquartile range and minimum and maximum. (**a**) TGF-β production (*n* = 6–8). (**b**) VEGF production (*n* = 5). (**c**) Angiogenin production (*n* = 7). * significant at <0.05.

## Data Availability

The data presented in this study are available on request from the corresponding author.

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
