# Peer review of "Human Macrophages Polarized by Interaction with Apoptotic Cells Produce Fibrosis-Associated Mediators and Enhance Pro-Fibrotic Activity of Dermal Fibroblasts In Vitro"

_cells, 2023, doi:10.3390/cells12151928_

Round 1

Reviewer 1 Report

Summary:

In this original article, Maksimova et al., studied the M2(LS) ability to produce fibrosis-associated factors, and the influence on proliferation, differentiation, and collagen-producing activity of primary dermal fibroblasts compared to classically and alternatively activated human macrophages. Overalll, the manuscript is well written and the topic is of great relevance. However, I have some questions and suggestions that may help to improve the quality of the paper.

- In the methodology, the authors did not add references. If the methods used in the study have already been performed in previous studies, please add the reference

- The authors analyzed the expression of MMPs (MMP2 and MMP9). It would be interesting to observe the activity of MMPs in zymography. With this experiment, other active proteases could be monitored in the study.

- For murine macrophages, it is known that CD86 and HLA-DR are good markers to monitor M1 polarization. In the case of human macrophages, these two markers are considered reliable.

- Many research results suggest that polyamines participate in cell proliferation, differentiation, and the regulation of gene expression, including genes that encode pro-fibrotic proteins. The authors could, if possible, evaluate such a phenomenon, or talk about this important topic in the discussion section.

In my opinion, the manuscript requires minor editing of English language before publication.

Author Response

Thank you very much for your good comments and suggestions. We have carefully reviewed the comments and have revised the manuscript accordingly. In accordance with your comment about english language, we have corrected typos and misspelled words. Changes to the manuscript are shown in yellow.

Our responses are given in a point-by-point manner below.

Point 1: In the methodology, the authors did not add references. If the methods used in the study have already been performed in previous studies, please add the reference

Response 1: Thank you for your suggestion. We added references in the Materials and Methods.

Point 2: The authors analyzed the expression of MMPs (MMP2 and MMP9). It would be interesting to observe the activity of MMPs in zymography. With this experiment, other active proteases could be monitored in the study.

Response 2: MMPs production was one of the small fragments of our study, and we did not focus specifically on it because it is a very complex issue that requires a particular analysis. We plan to investigate this issue in more detail later, and we will study protease activity in zymography.

Point 3: For murine macrophages, it is known that CD86 and HLA-DR are good markers to monitor M1 polarization. In the case of human macrophages, these two markers are considered reliable.

Response 3: Unlike murine macrophages, human macrophages cannot be characterized only phenotypically, and M1 and M2 markers can be expressed by both M1 and M2 macrophages depending on polarization conditions. For example, CD86 and HLA-DR are highly expressed on М2b macrophages (Wang LX, et al., 2019. DOI: 10.1002/JLB.3RU1018-378RR). Beyer M, et al. showed that human M-CSF-differentiated M1 and M2 macrophages did not differ in CD86 expression, and M-CSF and GM-CSF-differentiated M1 and M2 had the same level of HLA-DR expression (DOI: 10.1371/journal.pone.0045466). That’s why we believe that allostimulatory activity helps a lot for M1/M2 phenotype definition.

Point 4: Many research results suggest that polyamines participate in cell proliferation, differentiation, and the regulation of gene expression, including genes that encode pro-fibrotic proteins. The authors could, if possible, evaluate such a phenomenon, or talk about this important topic in the discussion section.

Response 4: A lot of mediators are involved in the regulation of fibrosis; the description of everyone would turn the original article into a review. We tried to discuss exactly the data that we received.

Reviewer 2 Report

1.     It is an interesting study. From it, we can understand the characters of human M1-like, M2-like macrophages induced by LPS, or IL-4, or LS, and the effects of the polarized macrophages on human dermal fibroblast functions in vitro. However, I didn't see any studies on the " polarization via interaction with apoplectic cells”, which was mentioned in the title of manuscript, the last paragraph in Result, and the sentence at Line 318-321 in Discussion. What are the apoptotic cells in this study, and did you proof them?

2.     NAF1 used in the study are primary dermal fibroblasts isolated from 1 burn patient, but not a stable cell line. Because of a big individual difference, do you think the cell behavior from this patient can represent all human dermal fibroblasts?

3.     Do you have M1-like macrophages induced by another reagent? if not, the "LPS" after "M1" is not necessary.

4.     You collected the blood samples from 72 donors, and each experiment number is about 4-16. It means you use M1-like or M2-like macrophages from different patients in different experiments. If it is true, how did you choose your samples for each experiment?

Author Response

Thank you very much for your good comments and suggestions. We have carefully reviewed the comments and have revised the manuscript accordingly. Changes to the manuscript are shown in yellow.

Our responses are given in a point-by-point manner below.

Point 1: It is an interesting study. From it, we can understand the characters of human M1-like, M2-like macrophages induced by LPS, or IL-4, or LS, and the effects of the polarized macrophages on human dermal fibroblast functions in vitro. However, I didn't see any studies on the " polarization via interaction with apoplectic cells”, which was mentioned in the title of manuscript, the last paragraph in Result, and the sentence at Line 318-321 in Discussion. What are the apoptotic cells in this study, and did you proof them?

Response 1: We obtained apoptotic cells by induced apoptosis in the fraction of non-adhesive mononuclear cells via serum deficiency as described in Materials and Methods. Efferocytosis (interaction with apoptotic cells) significantly affects macrophage polarization (Zizzo G, et al., 2012. DOIi: 10.4049/jimmunol.1200662; Gordon S, et al., 2018, DOI: 10.3389/fimmu.2018.00127; Han M, et al, 2021. DOI: 10.3390/life11111141). We have looked at this issue for a long time because efferocytosis plays a key role in the development of many diseases and is the main "switch" of the M1/M2 macrophage phenotype (Sakhno LV, et al. 2016, DOI: 10.1111/sji.12401; Chernykh ER, et al. 2018, DOI: 10.1016/j.cellimm.2018.06.002; Sakhno LV, et al., 2019, DOI: 10.1007/s10517-019-04616-8). The importance of contact with apoptotic cells for macrophage phenotype is described in our article: Sakhno L.V., Shevela E.Ya., Tikhonova M.A., Ostanin A.A., Chernykh E.R. Influence of deprivation apoptosis on gm-csf-induced macrophage differentiation (in rus). Immunologiya. 2017; 38(2):87-90. DOI: 10.18821/0206-4952-2017-38-2-87-90.

Point 2: NAF1 used in the study are primary dermal fibroblasts isolated from 1 burn patient, but not a stable cell line. Because of a big individual difference, do you think the cell behavior from this patient can represent all human dermal fibroblasts?

Response 2: NAF1 are not primary dermal fibroblasts, NAF1 is precisely the cell line obtained as a result of sequential passaging of primary dermal fibroblasts. Cell line is well described and used for modern research, for example, Kondrashov EV, et al., 2023 (DOI: 10.1016/j.bioorg.2023.106644).

Point 3: Do you have M1-like macrophages induced by another reagent? if not, the "LPS" after "M1" is not necessary.

Response 3: Since diverse mediators have been used alone or in various combinations to generate polarized macrophage populations, it was proposed describe macrophage activation stimulus in brackets (Murray P.J., et al., 2014; DOI: 10.1016/j.immuni.2014.06.008). In our work, we adhere to these recommendations for a clearer understanding of the macrophage generation protocol by readers. However, we have retained the M1/M2 nomenclature since it is still the most commonly used in the scientific community.

Point 4: You collected the blood samples from 72 donors, and each experiment number is about 4-16. It means you use M1-like or M2-like macrophages from different patients in different experiments. If it is true, how did you choose your samples for each experiment?

Response 4: Yes, we used macrophages from different donors for different experiments, although sometimes macrophages from a one donor have been analyzed in several tests (e.g., phenotype and the effect of macrophage conditioned media on the fibroblast differentiation, or allostimulatory activity and production of growth factors and cytokines, etc.). Donors were randomly selected for each experiment and comparable for sex and age. We also want to note that it was a fundamental point for us that all macrophage subtypes were obtained from one donor. It allowed us to compare different macrophage subtypes. In addition, in order to assess the level of the MMP/TIMP ratio, we analyzed the content of factors in the supernatants from one donor.

Round 2

Reviewer 1 Report

Thanks to the authors for improving the quality of the manuscript. All my concerns were clarified.